# Accumulation of Genetic and Epigenetic Alterations in the Background Liver and Emergence of Hepatocellular Carcinoma in Patients with Non-Alcoholic Fatty Liver Disease

**DOI:** 10.3390/cells10113257

**Published:** 2021-11-21

**Authors:** Satoru Hagiwara, Naoshi Nishida, Kazuomi Ueshima, Yasunori Minami, Yoriaki Komeda, Tomoko Aoki, Masahiro Takita, Masahiro Morita, Hirokazu Chishina, Akihiro Yoshida, Hiroshi Ida, Masatoshi Kudo

**Affiliations:** Department of Gastroenterology and Hepatology, Kindai University Faculty of Medicine, Osakasayama 589-8511, Japan; hagi-318@hotmail.co.jp (S.H.); kaz-ues@med.kindai.ac.jp (K.U.); minkun@med.kindai.ac.jp (Y.M.); y-komme@mvb.biglobe.ne.jp (Y.K.); t.aoki1918@gmail.com (T.A.); masahirot2797@yahoo.co.jp (M.T.); s0750081@yahoo.co.jp (M.M.); chocolatecake3candy@yahoo.co.jp (H.C.); ackiy0409@gmail.com (A.Y.); hidakuhp@gmail.com (H.I.); m-kudo@med.kindai.ac.jp (M.K.)

**Keywords:** genome, liver cancer, mutation, non-alcoholic fatty liver disease, methylation

## Abstract

The incidence of hepatocellular carcinoma (HCC) related to non-alcoholic fatty liver disease (NAFLD) is increasing worldwide. We analyzed 16 surgically resected HCC cases in which the background liver was pathologically diagnosed as NAFLD. Specimens with Brunt classification grade 3 or higher were assigned as the fibrotic progression group (*n* = 8), and those with grade 1 or lower were classified as the non-fibrosis progression group (*n* = 8). Comprehensive mutational and methylome analysis was performed in cancerous and noncancerous tissues. The target gene mutation analysis with deep sequencing revealed that CTNNB1 and TP53 mutation was observed in 37.5% and TERT promoter mutation was detected in 50% of cancerous samples. Furthermore, somatic mutations in non-cancerous samples were less frequent, but were observed regardless of the progression of fibrosis. Similarly, on cluster analysis of methylome data, status for methylation events involving non-cancerous liver was similar regardless of the progression of fibrosis. It was found that, even in cases of non-progressive fibrosis, accumulation of gene mutations and abnormal methylation within non-cancerous areas were observed. Patients with NAFLD require a rigorous liver cancer surveillance due to the high risk of HCC emergence based on the accumulation of genetic and epigenetic abnormalities, even when fibrosis is not advanced.

## 1. Introduction

Non-alcoholic fatty liver disease (NAFLD) is a phenotype of metabolic syndrome involving the liver, with disease severity ranging from non-alcoholic fatty liver to NAFLD. It is a disease concept that covers a wide range of liver pathologies that progress to liver cirrhosis and hepatocellular carcinoma (HCC) [1,2]. In Japan, non-B non-C (NBNC) HCC, which is negative for HBs’ antigen and negative for HCV antibody, has been on the rise in recent years. Among them, NAFLD-related liver cancer (NAFLD-HCC) derived from NAFLD against the background of lifestyle-related diseases is increasing [3,4,5,6,7].

Background inflammation of the liver and progression of fibrosis are extremely involved in the development of hepatocellular carcinoma. Therefore, studying genetic or epigenomic changes in the background liver may be useful in identifying patients with hepatic carcinogenesis. In patients with hepatitis B or C, carcinogenesis due to advanced cirrhosis is common, but carcinogenesis due to non-progressive cirrhosis is found in approximately 30–40% of cases of NAFLD-related HCC [8]. Therefore, carcinogenesis from mild fibrotic cases is commonly observed in patients with NAFLD-HCC. In clinical practice, it is currently very difficult to diagnose liver carcinogenesis from patients who do not have such progress of fibrosis at an early stage, and it is considered an urgent task to retain patients. However, the background conditions for the degree of accumulation of genetic and epigenetic alterations in non-cancerous liver have not been clarified.

In this study, we investigated the relationship between the progression of fibrosis in background livers as well as tumor tissues with NAFLD-HCC and the accumulation of genetic and epigenetic alterations in the affected patients.

## 2. Materials and Methods

### 2.1. Patients

Among the HCC patients who underwent surgery at Kindai University Hospital after October 2006, 77 patients who were diagnosed as NAFLD pathologically in background liver Alcohol consumption were examined using the questionnaire survey on the patients’ medical records. The cases with alcohol consumption of more than 30 g/day for men and 20 g/day for women were excluded from the study. Of them, 32 HCV antibody-positive patients, 15 HBs’ Ag-positive patients, and 10 HBc antibody-positive patients were excluded from further analysis. Among the remaining 20 patients who were negative for all hepatitis virus-related markers, eight patients with a background liver of Brunt classification stage 3 or higher were considered as the severe fibrosis group. Eight cases were Brunt classification stage 1 or lower and classified into the mild fibrosis group (Appendix A). The stage of liver fibrosis in the background liver was evaluated with the surgical specimens for HCC.

Mutational and methylation analysis was performed on these 16 cases.

### 2.2. Diagnosis of Metabolic Syndrome

As an essential item, the waist diameter should be 85 cm or more for men and 90 cm or more for women or the visceral fat area should be 100 cm^2^ or more for both men and women. Next, as a selection item, patients who met two or more of the three items were diagnosed with metabolic syndrome.

(1)Serum triglyceride is 150 mg/dL or more or serum high-density lipoprotein cholesterol is less than 40 mg/dL.(2)Systolic blood pressure is 130 mmHg or higher or diastolic blood pressure is 85 mmHg or higher.(3)Fasting blood glucose is 110 mg/dL or higher.

### 2.3. Pathological Diagnosis of Non-Alcoholic Steatohepatitis (NASH)

We calculated the NAFLD activity score (NAS) for the diagnosis of NASH [9]. The Brunt classification was also applied to evaluate the severity of NASH [10].

### 2.4. Ethics

The study protocol conformed to the ethical guidelines of the 1975 Declaration of Helsinki and was approved by the Institutional Review Board of Kindai University Faculty of Medicine (Approval Number 25-044). Written, informed consent was obtained from all patients recruited in the study.

### 2.5. Detection of Somatic Mutations

Frozen tissue was used for mutation analysis. When collecting non-cancerous tissue from frozen specimens, it was collected at a distance of at least 5 cm. Furthermore, it was confirmed by HE staining that there was no infiltration of liver cancer in non-cancerous tissues. The target sequence was sequenced using an Ion Proton sequencer. The HCC and liver tissues were stored at −80 °C until the extraction of DNA. Mutational analysis of known cancer driver genes was performed using the Ion AmpliSeq™ Comprehensive Cancer Panel (Illumina^®^, San Diego, CA, USA). Mutation of the telomerase reverse transcriptase (TERT) promoter was analyzed, as previously reported [11]. For the annotation of the target genes, we used CLC Genomics Workbench 9 (QIAGEN, Venlo, The Netherlands). After non-synonymous mutations were selected and low-quality sequences were excluded. Mutations in the human genome variation database (HGVD) version 2.3 and those observed in matched non-cancerous liver tissue were also filtered out. Allele reads were counted, and tissues were considered to contain clonally expanded mutant cells if the number of corresponding absolute mutant alleles was 20 leads or more. We also deferred to two types of prediction algorithms to determine the effect of missense mutations on carcinogenesis: the Sorting Intolerant From Tolerant (SIFT) functional prediction algorithm and the Polymorphism Phenotyping v2 (PolyPhen-2) functional prediction algorithm. The mutations that were classified as “tolerated” in SIFT functional prediction and “benign” in PolyPhen-2 functional prediction were excluded from further analyses.

### 2.6. Detection of Epigenome Mutations

#### 2.6.1. Comprehensive Analysis of DNA Methylation Level

Comprehensive methylation analysis of frozen tissue was performed using Human Methylation 450 Bead Array (HM450; Illumina, San Diego, CA, USA). First, 65 reference Probes were excluded from the probe containing CpG at 480,000 sites. The probe on the sex chromosome was excluded to avoid bias due to gender differences. Then, dbSNPs that included SNPs with a minor allele frequency of 5% or more were also excluded. We calculated the β value that represented the methylation levels as the ratio of signal intensity of the methylated allele divided by the sum of signal from the unmethylated and methylated allele + 100. Each β value was accompanied by a detection *p*-value, which indicated statistical significance against the background. Only β values with a detection *p*-value of >0.05 were included in the data analysis, as reported previously [12]. Multiple tests were performed using samples from the liver cancer patients and (1) CpG sites with an increase of the methylation level in the cancerous region (difference in average methylation level ≥35%) compared to the non-cancerous part and (2) those located around the transcription start point. The CpG site 1500 bp upstream of the translation start point was extracted from the analysis. Finally, 378 CpG sites satisfied these conditions [13].

#### 2.6.2. Confirmation of Expression Recovery by Pharmacological Unmasking

Eleven genes matching the 378 genes extracted from the comprehensive analysis of DNA methylation and 689 genes from pharmacological unmasking were selected for further analysis. The 11 genes extracted were TTBK1, FOXL2, PPP1R14A, SPINT2, NPTX2, CHGA, BVES, EFNB2, BMP4, DKK3, and TM6SF1.

For a control, we used the methylation level of simple fatty liver (non-alcoholic fatty liver (NAFL)) without HCC determined by HM450 BeadChip. These methylation data were obtained from the NCBI-GEO database (Dataset: GSE48325).

### 2.7. Statistical Analysis

Principal component analysis was used to stratify the methylation profile. In the hierarchical cluster analysis, Fisher’s exact test was used to make a comparison between groups.

For the comparative study of fibrosis progression and methylation level, Z score was calculated. Comparisons of the methylation level among the groups were performed using Dunnett’s test and Tukey–Kramer’s HSD test. All statistical analyses were performed using the SPSS software (version 11.5; SPSS Inc., Chicago, IL, USA).

## 3. Results

We analyzed 16 HCC cases in which the background liver was pathologically diagnosed as NAFLD via surgical specimens. Those that were Stage 3 and above of the Brunt classification were classified into the severe fibrosis group (*n* = 8), and stage 1 and below were classified into the mild fibrosis development group (*n* = 8). Clinical and pathological results were compared for age, sex, body mass index (BMI), and frequency of diabetes mellitus (DM) and hypertension (HT) complications; no significant differences were observed for the background between the two groups. Serum alanine aminotransferase (ALT) levels were higher in the severe fibrosis group (*p* = 0.19), although no significant difference was observed. Platelet (PLT), on the other hand, was lower in the severe fibrosis group, although no significant difference was observed (*p* = 0.08). There was no statistical difference for the maximum tumor diameter, number of tumors, serum α-fetoprotein (AFP) value, and Des-Gamma-Carboxy prothrombin (DCP) value between the two groups. Regarding the NAS score, intralobular inflammation was significantly higher in the severe fibrosis group (*p* < 0.01). Regarding the Brunt classification, both grade and stage were significantly higher in the severe fibrosis group (*p* = 0.001 and *p* < 0.001, respectively) (Table 1).

The somatic gene mutations were detected exclusively in the HCC for 24 genes and totaled 44 mutations in the severe fibrosis group, while there were 21 genes that totaled 39 mutations in the mild fibrosis group. The number of somatic gene mutations detected in the non-cancerous liver as well as in the HCC was eight genes in the severe fibrosis group, totaling 11 mutations, and seven genes totaling 11 mutations in the mild fibrosis group (Table 2).

Somatic mutations in HCC were frequently observed regardless of degree of fibrosis in the background liver; CTNNB1 mutation was detected in 37.5%, TP53 mutation was 37.5%, and TERT promoter mutation was 50% in the HCC. On the other hand, somatic mutations in non-cancerous liver were less frequent than those in HCC, but these were also observed at a constant frequency regardless of degree of fibrosis (Figure 1).

Principal component analysis using methylation level was performed on 11 target genes using cancerous, non-cancerous, and simple fatty liver (control) samples. The methylation profile in the HCC was different from that found in the non-cancerous liver and control. However, methylation profiles in the non-cancerous liver and control were similar. The contributions of each principal component were 80.5% for CP1 80.5%, 6.8% for CP2, and 4.4% for CP3 (Figure 2).

Next, principal component analysis of methylation levels in the 11 genes was performed for non-cancerous background liver of severe fibrosis and mild fibrosis groups and control, simple fatty liver samples. Although, the methylation profiles of the background liver from severe fibrosis and that from mild fibrosis tended to overlap, these were different from the methylation profiles in simple fatty liver (control) (Figure 3).

Hierarchical cluster analysis was performed to clarify the differences in methylation levels between the background liver from the severe fibrosis and mild fibrosis groups. As expected, the methylation levels were higher in the liver of severe fibrosis and mild fibrosis than those in the control, simple fatty livers (*p* = 0.0007). In contrast, no statistical difference was observed for the methylation levels between the background liver from severe fibrosis and mild fibrosis in HCC patients (Figure 4).

A comparison of the methylation level was performed using a Z score of the 11 genes to clarify the difference in methylation levels of the background liver of HCC patients between severe fibrosis and mild fibrosis. The methylation levels of these two groups were higher than those of the control, simple fatty liver (Figure 5).

## 4. Discussion

In recent years, rates of NAFLD-related liver cancer (NAFLD-HCC), based on obesity and diabetes, have been increasing [3,4,5,6]. Carcinogenesis from mild fibrotic cases is often observed in patients with NAFLD-HCC [8], but the degree of accumulation of somatic mutations and epigenome mutations in the liver is unknown.

Genetic abnormalities within the cancerous part of NAFLD-related liver have been reported previously by Kim et al. A mutation in the TERT promoter was found in 82% of the cases with NAFLD-related liver cancer, and chromosome 8p deletion was observed in all cases [14]. In addition, Pinyol et al. reported that in 80 cases of NAFLD-related liver cancer mutations in the TERT promoter were the most common, seen in 56% of cases, similar to the report by Kim et al. In addition, 28% had *CTNNB1*, 18% had *TP53*, and 10% had *ACVR2A* mutations [15]. In our study, the TERT promoter mutation was the most commonly observed, in 50%, followed by *CTNNB1* and *TP53* mutations in 37.5%. In this study, we investigated the accumulation of gene mutations in the background liver of NAFLD-HCC patients. In particular, we examined whether progression of fibrosis was related to the accumulation of mutations. Several gene mutations were detected in the background liver from severe fibrosis and mild fibrosis. In other words, in NAFLD-HCC, gene mutations were accumulated not only within the cancerous part but also within the non-cancerous liver, regardless of the progression of liver fibrosis.

Many studies on cancerous and non-cancerous tissues have reported abnormal methylation. In studies involving cancerous and non-cancerous areas, abnormal methylation of the *TRIM4*, the *PRC1* gene, the *WHSC1*, and the *MAML3* genes was found within cancerous areas. Studies involving the cancerous and non-cancerous regions found abnormal methylation of the TRIM4, PRC1, WHSC1, and MAML3 genes within the cancerous region. These have been reported as methylations characteristic of NAFLD-related liver cancer [16,17]. In addition, previous studies show the unique methylation events in the *FGFR2*, the *CASP1*, the *PPARα*, the PPARƔ, and the *MT-ND6* genes in the liver with progression of fibrosis [18,19,20]. Previously, we reported that number of methylation events in the liver of hepatitis C patients is positively associated with the shorter time to HCC emergence [21]. Therefore, it is conceivable that the alteration of genes in the background liver should be a risk of HCC emergence even for the cases with NAFLD. In our study, principal component analysis using methylation levels was performed on the 11 genes we targeted, using cancerous and non-cancerous liver of HCC and simple fatty liver (control) samples. The methylation profile in the cancerous part was different from that in the non-cancerous part and control, but it was similar among the livers in the non-cancerous part. In order to investigate methylation abnormalities due to the progress of fibrosis, we analyzed methylation events of the 11 genes in non-cancerous livers with sever fibrosis, those with mild fibrosis, and control (simple fatty liver) using principal component analysis and hierarchical clustering analysis. The methylation profiles of the background livers from sever fibrosis and mild fibrosis were different than those of simple fatty liver (control), whereas the methylation profiles of sever fibrosis and mild fibrosis tended to overlap each other. The degree of gene mutations and aberrant methylation were similar in patients with emergence of HCC regardless of the degree of fibrosis.

In this study, we found that in patients with hepatic carcinogenesis, mutations and abnormal methylation had already accumulated within the non-cancerous background liver, even in the patients with mild liver fibrosis. Therefore, Patients with NAFLD require a rigorous surveillance for the emergence of HCC, even when fibrosis is not advanced. This study has some limitations: (1) the number of cases examined was a small number, which may have led to a biased conclusion, (2) patients with HBV or HCV or alcoholic steatohepatitis patients were not examined for comparison with NAFLD, where specific alterations in the NAFLD liver cannot be detected, and (3) the Background liver in patients with NASH and simple fatty liver who do not develop HCC was not tested as a control. Further studies are still required to know the role of background alterations of genes for hepatocarcinogenesis in NAFLD liver. Nevertheless, by identifying the combination of gene mutations and abnormal methylation that most affects liver carcinogenesis in the future, we believe that more effective surveillance may be possible for the patients with NAFLD.

## 5. Conclusions

It was found that, even in cases with mild fibrosis of NAFLD, gene mutations and abnormal methylations were already accumulated in the background liver in patients with the emergence of HCC. Strict liver cancer surveillance is required because of the accumulation of genetic abnormalities and abnormal methylation even in such cases.

## Figures and Tables

**Figure 1 cells-10-03257-f001:**
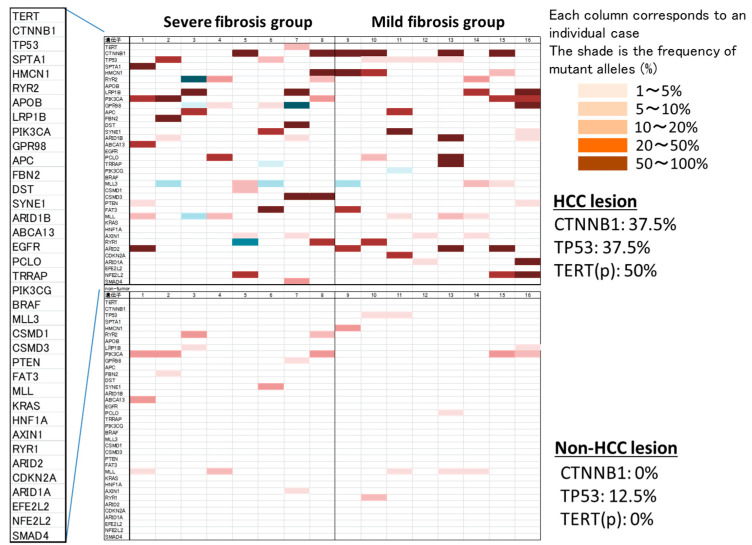
Frequency of somatic mutation detection in cancerous and non-cancerous areas by NSG (heat map). Somatic mutations in the HCC were frequently observed: 37.5% in the *CTNNB1* gene, 37.5% in the *TP53* gene, and 50% in TERT promoter. On the other hand, although somatic mutations in non-cancerous liver were less frequent, these were observed at a constant frequency regardless of degree of fibrosis.

**Figure 2 cells-10-03257-f002:**
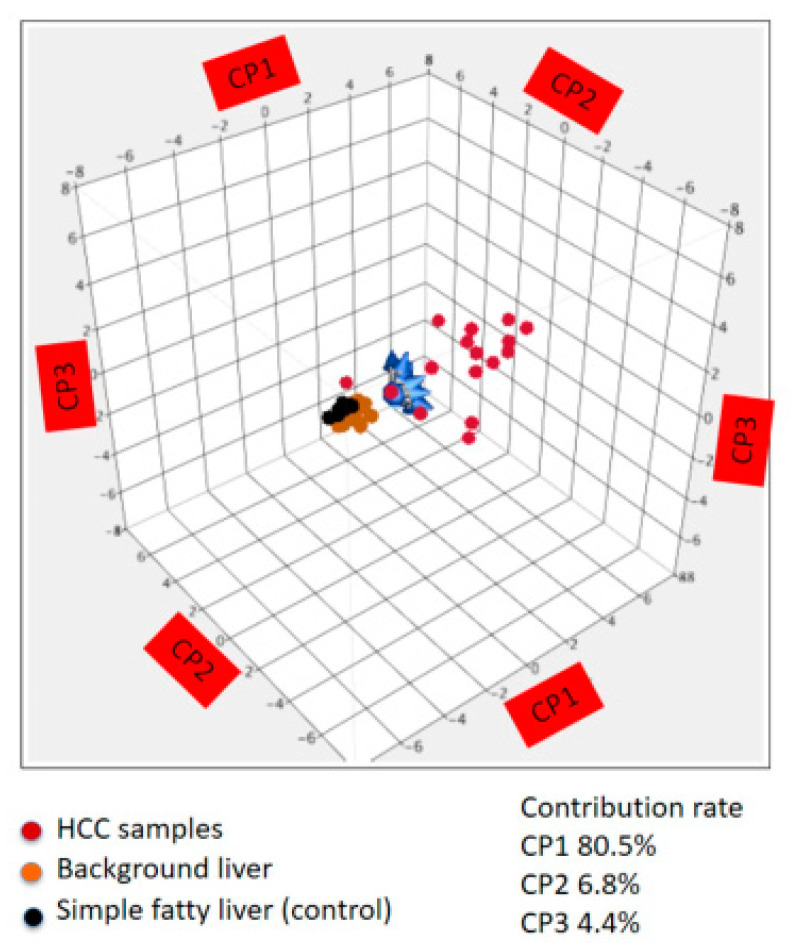
Principal component analysis using methylation level. HCC (*n* = 16) vs. non-cancerous background liver (*n* = 16) vs. simple fatty liver (control, *n* = 10). The methylation profile in the HCC was different from that in the non-cancerous background liver and simple fatty liver (control).

**Figure 3 cells-10-03257-f003:**
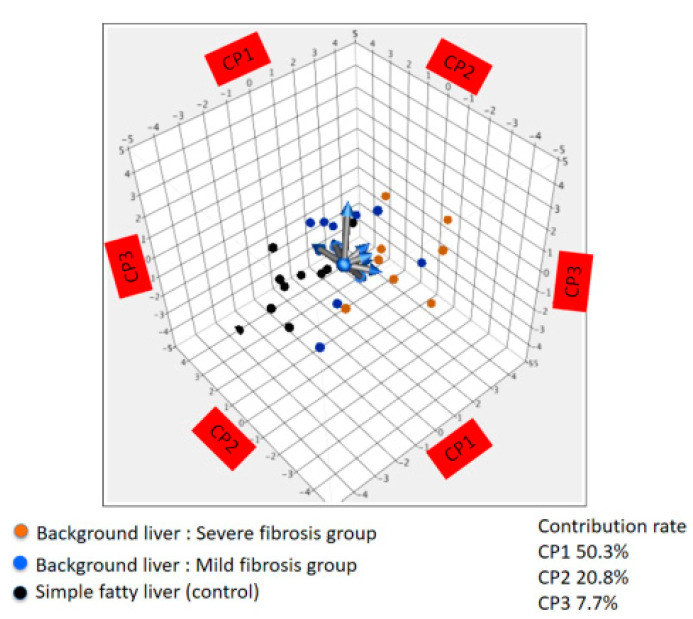
Principal component analysis using methylation level. Background livers with severe fibrosis (*n* = 8) vs. mild fibrosis (*n* = 8) vs. simple fatty liver (control, *n* = 10). The methylation profiles from the background livers with severe fibrosis and mild fibrosis tended to overlap. The contributions of each principal component were CP1 (50.3%), CP2 (20.8%), and CP3 (7.7%).

**Figure 4 cells-10-03257-f004:**
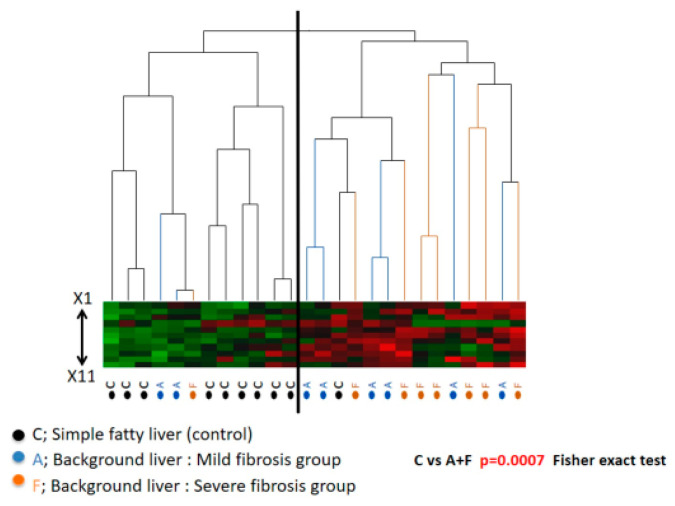
Hierarchical cluster analysis using methylation level. The methylation levels in the background liver of severe fibrosis and mild fibrosis were higher than those of simple fatty liver (control) (*p* = 0.0007). On the other hand, the methylation levels in the background liver of severe fibrosis and mild fibrosis were similar.

**Figure 5 cells-10-03257-f005:**
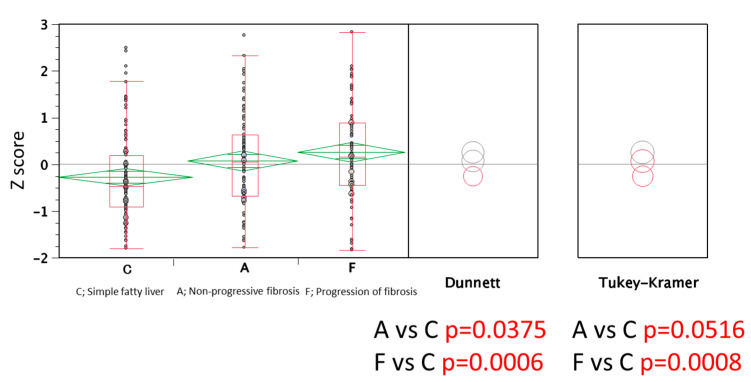
Relationship between progression of fibrosis and methylation levels in non-cancerous liver tissue. Methylation levels in the background liver from severe fibrosis and those from mild fibrosis were higher than those in the control. In contrast, the methylation levels in severe fibrosis and mild fibrosis were similar.

**Table 1 cells-10-03257-t001:** Comparison of clinical and pathological results of patients.

	Severe Fibrosis Group (*n* = 8)	Mild Fibrosis Group (*n* = 8)	*p*-Value
Age (year, mean ± SD)	71 ± 6	70 ± 6	0.68
Sex Male-no. (%)	6 (75)	6 (75)	1
B.M.I. (mean ± SD)	25 ± 3.0	25 ± 2.7	0.69
Diabetes complications -no. (%)	3 (37.5)	6 (75)	0.31
Hypertension complications -no. (%)	3 (37.5)	4 (50)	1
Metabolic syndrome -no. (%)	3 (37.5)	3 (37.5)	1
ALT (IU/L, mean ± SD)	51 ± 27	35 ± 17	0.19
ALB (g/dL, mean ± SD)	3.9 ± 0.2	4.1 ± 0.6	0.38
PLT (×104/μL, mean ± SD)	16 ± 4.3	20 ± 4.2	0.08
Maximum tumor diameter (mm, mean ± SD)	46 ± 25	76 ± 45	0.11
Single tumor number -no. (%)	8 (100)	7 (88)	1
AFP (ng/mL, median, range)	7 (4–65)	4.5 (2–2044)	0.44
DCP (mAU/mL, median, range)	226 (15–52,788)	673 (14–44,716)	0.79
NAS (mean ± SD)	4.6 ± 1.1	3.6 ± 1.1	0.09
Steatosis	1.3 ± 0.4	1.3 ± 0.4	1
Lobular inflammation	1.9 ± 0.3	1.1 ± 0.3	<0.01
Ballooning	1.5 ± 0.5	1.3 ± 0.7	0.43
Brunt classification -grade (mean ± SD)	2.0 ± 0.5	1.1 ± 0.3	0.001
Brunt classification -stage (mean ± SD)	3.6 ± 0.5	0.8 ± 0.4	<0.001

BMI, body mass index; ALT, alanine aminotransferase; AFP, α-fetoprotein; DCP, Des-Gamma-Carboxy prothrombin; NAS, NAFLD activity score.

**Table 2 cells-10-03257-t002:** Detection of somatic mutations in cancerous and non-cancerous areas by a next-generation sequencer (NGS).

	Severe Fibrosis Group	Mild Fibrosis Group
NT (-) ⇒ HCC (+) Somatic gene mutations that did not exist in the non-cancerous part but existed only in the cancerous part	GPR98 (6) MLL3 (5) RYR1 (3), MLL (3), TP53 (3), TERTpromoter (3) CTNNB1 (2), CSMD3 (2), ARID1B (2) SPTA1, PTEN, ARID2, APC, RYR2 PCLO, CSMD1, AXIN1, NFE2L2, TERT TRRAP, FAT3, LRP1B, DST, SMAD4	TERTpromoter (5) CTNNB1 (4) MLL3 (3), ARID2 (3) HMCN1 (2), CYNE1 (2) TP53 (2), AXIN1 (2) ARID1A (2), ARID1B (2) NFE2L2 (2) FAT3, PCLO, APC PIK3CG, CDKN2A TPRAP, RYR2, LRP1B GRP98, PTEN
NT (+) ⇒ HCC (++) Somatic gene mutation that was present in the non-cancerous part and also in the cancerous part	PIK3CA (3) MLL (2) ABCA13, FBN2, RYR2, LRP1B, SYNE1 AXIN1	MLL (3) TP53 (2), PIK3CA (2) HMCN1, RYR1, PCLO LRP1B

NT, non-cancerous liver; HCC, Hepatocellular Carcinoma.

## Data Availability

The data presented in this study are available on request from the corresponding author.

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
