# Peer review of "Accumulation of Genetic and Epigenetic Alterations in the Background Liver and Emergence of Hepatocellular Carcinoma in Patients with Non-Alcoholic Fatty Liver Disease"

_cells, 2021, doi:10.3390/cells10113257_

Round 1

Reviewer 1 Report

The manuscript entitled “Accumulation of Genetic and Epigenetic Alterations in the Background Liver and Emergence of Hepatocellular Carcinoma in Patients with Non-Alcoholic Fatty Liver Disease” by Satoru Hagiwara, et al. reported that the relationship between the progression of fibrosis in background livers as well as tumor tissues with NAFLD-HCC and the accumulation of genetic and epigenetic alterations in the affected patients, and they claimed that strict liver cancer surveillance is required because of the accumulation of genetic abnormalities and abnormal methylation even in cases with mild fibrosis. Although the study is of interest and important, there are some flaws to be revealed.

Major comments

  1. The authors should create the flowchart of the patients included in this study for good understanding at the section “1. Patients”.
  2. Authors revealed that gene mutations were accumulated not only within the cancerous part but also within the non-cancerous liver, regardless of the progression of liver fibrosis in NAFLD-HCC. How is the difference of gene mutations in peripheral blood? It is very important the relationship between intrahepatic factors and those of peripheral blood.
  3. As the authors mentioned that background liver of NASH patients who do not develop HCC are not examined as a control as a limitation in this study, is there any impact of the existence of HCC for the difference of methylation profiles of the background livers between those from severe or mild fibrosis and those from simple fatty liver?
  4. The authors mentioned that accumulation of genetic and epigenetic alterations may take place probably due to the strong oxidative stress within a short period of time even in patients with mild fibrosis in Line 245-247, but I think this sentence was unnecessary because this statement is groundless and there is a leap in logic.

Minor comments

  1. The authors should correct “non-B non-C (NBNC) l HCC” to “non-B non-C (NBNC) HCC” in Line 31.
  2. The authors should unify “Severe fibrotic group” and “Mild fibrosis group” in Table 1.
  3. The authors should spell out “NSG” the first time in this manuscript. It is hard to understand in Table 2 or Figure 1.
  4. The authors should describe the number each Figure in order of appearance and cite them by number in the manuscript, for example (Figure 2) in Line 169, (Figure 3) in Line 178, so on.

Author Response

Point-by-Point Response to the Editors and Reviewers comments:

We appreciate the reviewer’s positive comments to our manuscript. We have studied his/her comments carefully and addressed the queries as below.

Reviewer 1’s comment:

The manuscript entitled “Accumulation of Genetic and Epigenetic Alterations in the Background Liver and Emergence of Hepatocellular Carcinoma in Patients with Non-Alcoholic Fatty Liver Disease” by Satoru Hagiwara, et al. reported that the relationship between the progression of fibrosis in background livers as well as tumor tissues with NAFLD-HCC and the accumulation of genetic and epigenetic alterations in the affected patients, and they claimed that strict liver cancer surveillance is required because of the accumulation of genetic abnormalities and abnormal methylation even in cases with mild fibrosis. Although the study is of interest and important, there are some flaws to be revealed.

Major comments

Queries 1:

The authors should create the flowchart of the patients included in this study for good understanding at the section “1. Patients”.

Reply;

Thank you for your valuable suggestions. I created a patient flowchart and added it as supplementary figure1.

Queries 2:

Authors revealed that gene mutations were accumulated not only within the cancerous part but also within the non-cancerous liver, regardless of the progression of liver fibrosis in NAFLD-HCC. How is the difference of gene mutations in peripheral blood? It is very important the relationship between intrahepatic factors and those of peripheral blood.

Reply;

Thank you for your valuable suggestions. This study does not compare with gene mutations in peripheral blood. However, CLC Genomics Workbench 9 is used for annotation of the target gene, and non-cynonymous mutation and low quality sequence are excluded. In addition, all mutations detected in the Human genome variation database (HGVD) version 2.3 have been excluded. In addition, the Sorting Intolerant From Tolerant (SIFT) function prediction algorithm and the Polymorphism Phenotyping v2 (PolyPhen-2) function prediction algorithm were used to determine the effect of missense mutations on carcinogenesis. Mutations classified as "acceptable" in SIFT function prediction and "benign" in PolyPhen-2 function prediction were excluded from further analysis. In this way, we make full use of various methods to detect somatic mutations. This content is specified in methods 2.5 (p2, line90 – p3, line101).

Queries 3:

As the authors mentioned that background liver of NASH patients who do not develop HCC are not examined as a control as a limitation in this study, is there any impact of the existence of HCC for the difference of methylation profiles of the background livers between those from severe or mild fibrosis and those from simple fatty liver?

Reply;

Thank you for your valuable suggestions. I think that the methylation analysis of the background liver in patients with simple fatty liver with hepatocellular carcinoma is a very important point.

However, liver carcinogenesis from patients with simple fatty liver is extremely rare and could not be analyzed in this study. I would like to increase the number of cases in the future and consider it.

This content has been added as a limitation (p9, line265-266).

Queries 4:

The authors mentioned that accumulation of genetic and epigenetic alterations may take place probably due to the strong oxidative stress within a short period of time even in patients with mild fibrosis in Line 245-247, but I think this sentence was unnecessary because this statement is groundless and there is a leap in logic.

Reply;

Thank you for your valuable suggestions. As you pointed out, there is no basis. This sentence will be deleted.

Minor comments

Queries 1:

The authors should correct “non-B non-C (NBNC) l HCC” to “non-B non-C (NBNC) HCC” in Line 31.

Reply;

Thank you for your valuable suggestions. I have fixed it.

Queries 2:

The authors should unify “Severe fibrotic group” and “Mild fibrosis group” in Table 1.

Reply;

Thank you for your valuable suggestions. I have fixed it.

Queries 3:

The authors should spell out “NSG” the first time in this manuscript. It is hard to understand in Table 2 or Figure 1.

Reply;

Thank you for your valuable suggestions. We have spelled out.

Queries 4:

The authors should describe the number each Figure in order of appearance and cite them by number in the manuscript, for example (Figure 2) in Line 169, (Figure 3) in Line 178, so on.

Reply;

A; Thank you for your valuable suggestions. I quoted it.

Reviewer 2 Report

In this study, the authors employed various mutational and epigenetic (methylation)-based approaches to investigate the relationship between the progression of fibrosis from background to tumor (HCC) in patients with NAFLD.  The study is well designed and well executed and the manuscript is very well written.  However, few issues need to be addressed.   

Major concerns:

  1. The authors did not describe/discuss the approach they used to distinguish between background vs. cancerous lesions in resected NAFLD-HCC tissue samples (see Fig. 1 for example). When describing background liver of NAFLD-HCC, was there any degree of spread or infiltration of cancerous cells.  In other words, how can the authors be sure that the mutations and methylation events detected in non-cancerous tissues are not resulting from fibrotic or cancerous cells spreading into normal tissue.
  2. It is clear that the supposed progression from simple fatty liver to mildly fibrotic liver to severely fibrotic live (figures 2-5) [and by deduction to HCC] associates with significant quantitative methylome variations. However, it is not as easy to discern how such methylome variations (11 genes) can be interpreted as biomarkers of progression from one state to the next.  The authors need to clarify this central point.
  3. In figure 2, the authors showed that the methylation pattern in non-cancerous tissues is different from that of cancerous tissues. It is not clear how this can be used to justify the conclusion that transition from normal to HCC associates with the accumulation of methylation events?
  4. One aspect of the data that was not adequately addressed is that the transition from normal to cancerous tissue in liver seems to have a quantitative component besides the qualitative one (methylation of specific genes) (see Fig. 5). The limitation here is that only 11 genes were examined.  It would be beneficial to expand the list of genes being assessed for methylation to see if this holds true.  This could also help in resolving point #3 above.       

Minor points:

  • Introduction is too short, can be stretched a bit to be more informative to the reader
  • In figure 1, the authors noted that each row corresponds to an individual case, please double check (is it row or column).
  • A table listing all altered genes (mutations or methylation) showing in HCC tissue samples as compared with counterparts in normal tissue samples along with their known roles in the initiation and/or progression of HCC will be very helpful for the reader and will help clarify multiple ambiguities in the results and discussion section.
  • Few typos/syntax errors need to be fixed. See for example discussion section, paragraph 2 line 12 (…. of the both groups …); paragraph 3 line 1 (…. reported case of abnormal…) paragraph 3 line 4 (…. event characteristic for …) etc.

Author Response

Point-by-Point Response to the Editors and Reviewers comments:

We appreciate the reviewer’s positive comments to our manuscript. We have studied his/her comments carefully and addressed the queries as below.

Reviewer 2’s comment:

In this study, the authors employed various mutational and epigenetic (methylation)-based approaches to investigate the relationship between the progression of fibrosis from background to tumor (HCC) in patients with NAFLD.  The study is well designed and well executed and the manuscript is very well written.  However, few issues need to be addressed.

Major concerns:

Queries 1:

The authors did not describe/discuss the approach they used to distinguish between background vs. cancerous lesions in resected NAFLD-HCC tissue samples (see Fig. 1 for example). When describing background liver of NAFLD-HCC, was there any degree of spread or infiltration of cancerous cells.  In other words, how can the authors be sure that the mutations and methylation events detected in non-cancerous tissues are not resulting from fibrotic or cancerous cells spreading into normal tissue.

Reply;

A; Thank you for your valuable suggestions. When collecting non-cancerous tissue from frozen specimens, it was collected at a distance of at least 5 cm. Furthermore, it has been confirmed by HE staining that there is no infiltration of liver cancer in non-cancerous tissues. It is specified in the text. This content is specified in the text (p2, line 83-86).

Queries 2:

It is clear that the supposed progression from simple fatty liver to mildly fibrotic liver to severely fibrotic live (figures 2-5) [and by deduction to HCC] associates with significant quantitative methylome variations. However, it is not as easy to discern how such methylome variations (11 genes) can be interpreted as biomarkers of progression from one state to the next.  The authors need to clarify this central point.

Reply;

A; Thank you for your valuable suggestions. The reported representative functions of the 11 genes analyzed are specified in supplementary table 3.

Queries 3:

In figure 2, the authors showed that the methylation pattern in non-cancerous tissues is different from that of cancerous tissues. It is not clear how this can be used to justify the conclusion that transition from normal to HCC associates with the accumulation of methylation events?

Reply;

A; Thank you for your valuable suggestions. As you pointed out, in Figure 2, the control and non-cancerous (n = 16) methylation profiles overlap. However, this result was obtained because the methylation profile of HCC was significantly different from that of the other two groups in this analysis. However, in the study excluding HCC (Figure 3), it was confirmed that the methylation profile was different between the control group and the NASH group.

Queries 4:

One aspect of the data that was not adequately addressed is that the transition from normal to cancerous tissue in liver seems to have a quantitative component besides the qualitative one (methylation of specific genes) (see Fig. 5). The limitation here is that only 11 genes were examined. It would be beneficial to expand the list of genes being assessed for methylation to see if this holds true.  This could also help in resolving point #3 above.

Reply;

Thank you for your valuable suggestions. The 378 genes extracted by analysis using the Human Methylation 450 Bead Array matched the 689 genes whose expression recovery was confirmed by pharmacological unmasking, and the methylation level could be determined in all cases. bottom. A list of methylation levels of 11 genes has been specified as supplementary table 3.

Minor points:

Queries 1:

Introduction is too short, can be stretched a bit to be more informative to the reader

Reply;

Thank you for your valuable suggestions. We have extended and enhanced the introduction.

Queries 2:

In figure 1, the authors noted that each row corresponds to an individual case, please double check (is it row or column).

Reply;

Thank you for your valuable suggestions. It has been modified from row to column.

Queries 3:

A table listing all altered genes (mutations or methylation) showing in HCC tissue samples as compared with counterparts in normal tissue samples along with their known roles in the initiation and/or progression of HCC will be very helpful for the reader and will help clarify multiple ambiguities in the results and discussion section.

Reply;

Thank you for your valuable suggestions. A list of mutations and methylations has been added as supplementary table 1.2.3.

Queries 4:

Few typos/syntax errors need to be fixed. See for example discussion section, paragraph 2 line 12 (…. of the both groups …); paragraph 3 line 1 (…. reported case of abnormal…) paragraph 3 line 4 (…. event characteristic for …) etc.

Reply;

Thank you for your valuable suggestions. Fixed typos / syntax errors.

Reviewer 3 Report

The topic of this paper is of great interest, and I consider the manuscript entitled “Accumulation of Genetic and Epigenetic Alterations in the Background Liver and Emergence of Hepatocellular Carcinoma  in Patients with Non-Alcoholic Fatty Liver Disease” suitable for publication at the present form.

Author Response

Point-by-Point Response to the Editors and Reviewers comments:

We appreciate the reviewer’s positive comments to our manuscript. We have studied his/her comments carefully and addressed the queries as below.

Reviewer 3’s comment:

The topic of this paper is of great interest, and I consider the manuscript entitled “Accumulation of Genetic and Epigenetic Alterations in the Background Liver and Emergence of Hepatocellular Carcinoma in Patients with Non-Alcoholic Fatty Liver Disease” suitable for publication at the present form.

Reply;

Thank you for your valuable feedback. Thank you for your evaluation.

We added some additional references based on the revised descriptions in the manuscript.

Round 2

Reviewer 1 Report

The authors carefully modified each revise comments  point-by-point.

I don't have any additional revise comments.